# A Stein variational Newton method

**Gianluca Detommaso**
University of Bath & The Alan Turing Institute
gd391@bath.ac.uk

**Tiangang Cui**
Monash University
Tiangang.Cui@monash.edu

**Alessio Spantini**
Massachusetts Institute of Technology
spantini@mit.edu

**Youssef Marzouk**
Massachusetts Institute of Technology
ymarz@mit.edu

**Robert Scheichl**
Heidelberg University
r.scheichl@uni-heidelberg.de

## Abstract

Stein variational gradient descent (SVGD) was recently proposed as a general purpose nonparametric variational inference algorithm [Liu & Wang, NIPS 2016]: it minimizes the Kullback–Leibler divergence between the target distribution and its approximation by implementing a form of functional gradient descent on a reproducing kernel Hilbert space. In this paper, we accelerate and generalize the SVGD algorithm by including second-order information, thereby approximating a Newton-like iteration in function space. We also show how second-order information can lead to more effective choices of kernel. We observe significant computational gains over the original SVGD algorithm in multiple test cases.

## 1   Introduction

Approximating an intractable probability distribution via a collection of samples—in order to evaluate arbitrary expectations over the distribution, or to otherwise characterize uncertainty that the distribution encodes—is a core computational challenge in statistics and machine learning. Common features of the target distribution can make sampling a daunting task. For instance, in a typical Bayesian inference problem, the posterior distribution might be strongly non-Gaussian (perhaps multimodal) and high dimensional, and evaluations of its density might be computationally intensive.

There exist a wide range of algorithms for such problems, ranging from parametric variational inference [4] to Markov chain Monte Carlo (MCMC) techniques [10]. Each algorithm offers a different computational trade-off. At one end of the spectrum, we find the parametric mean-field approximation—a cheap but potentially inaccurate variational approximation of the target density. At the other end, we find MCMC—a nonparametric sampling technique yielding estimators that are consistent, but potentially slow to converge. In this paper, we focus on Stein variational gradient descent (SVGD) [17], which lies somewhere in the middle of the spectrum and can be described as a particular *nonparametric* variational inference method [4], with close links to the density estimation approach in [2].

The SVGD algorithm seeks a deterministic coupling between a tractable reference distribution of choice (e.g., a standard normal) and the intractable target. This coupling is induced by a transport map $T$ that can *transform* a collection of reference samples into samples from the desired target distribution. For a given pair of distributions, there may exist infinitely many such maps [28]; several existing algorithms (e.g., [27, 24, 21]) aim to approximate feasible transport maps of various forms.

The distinguishing feature of the SVGD algorithm lies in its definition of a suitable map $T$. Its central idea is to approximate $T$ as a growing composition of simple maps, computed *sequentially*:

$$T = T_1 \circ \cdots \circ T_k \circ \cdots , \tag{1}$$

where each map $T_k$ is a perturbation of the identity map along the steepest descent direction of a functional $J$ that describes the Kullback–Leibler (KL) divergence between the pushforward of the reference distribution through the composition $T_1 \circ \cdots \circ T_k$ and the target distribution. The steepest descent direction is further *projected* onto a reproducing kernel Hilbert space (RKHS) in order to give $T_k$ a nonparametric closed form [3]. Even though the resulting map $T_k$ is available explicitly without any need for numerical optimization, the SVGD algorithm implicitly approximates a steepest descent iteration on a space of maps of given regularity.

A primary goal of this paper is to explore the use of second-order information (e.g., Hessians) within the SVGD algorithm. Our idea is to develop the analogue of a Newton iteration—rather than gradient descent—for the purpose of sampling distributions more efficiently. Specifically, we design an algorithm where each map $T_k$ is now computed as the perturbation of the identity function along the direction that minimizes a certain local quadratic approximation of $J$. Accounting for second-order information can dramatically accelerate convergence to the target distribution, at the price of additional work per iteration. The tradeoff between speed of convergence and cost per iteration is resolved in favor of the Newton-like algorithm—which we call a Stein variational Newton method (SVN)—in several numerical examples.

The efficiency of the SVGD and SVN algorithms depends further on the choice of reproducing kernel. A second contribution of this paper is to design geometry-aware Gaussian kernels that also exploit second-order information, yielding substantially faster convergence towards the target distribution than SVGD or SVN with an isotropic kernel.

In the context of *parametric* variational inference, second-order information has been used to accelerate the convergence of certain variational approximations, e.g., [14, 13, 21]. In this paper, however, we focus on *nonparametric* variational approximations, where the corresponding optimisation problem is defined over an infinite-dimensional RKHS of transport maps. More closely related to our work is the Riemannian SVGD algorithm [18], which generalizes a gradient flow interpretation of SVGD [15] to Riemannian manifolds, and thus also exploits geometric information within the inference task.

The rest of the paper is organized as follows. Section 2 briefly reviews the SVGD algorithm, and Section 3 introduces the new SVN method. In Section 4 we introduce geometry-aware kernels for the SVN method. Numerical experiments are described in Section 5. Proofs of our main results and further numerical examples addressing scaling to high dimensions are given in the supplementary material. Code and all numerical examples are collected in our GitHub repository [1].

## 2   Background

Suppose we wish to approximate an intractable target distribution with density $\pi$ on $\mathbb{R}^d$ via an empirical measure, i.e., a collection of samples. Given samples $\{x_i\}$ from a tractable reference density $p$ on $\mathbb{R}^d$, one can seek a transport map $T : \mathbb{R}^d \to \mathbb{R}^d$ such that the pushforward density of $p$ under $T$, denoted by $T_* p$, is a close approximation to the target $\pi$.[1] There exist infinitely many such maps [28]. The image of the reference samples under the map, $\{T(x_i)\}$, can then serve as an empirical measure approximation of $\pi$ (e.g., in the weak sense [17]).

**Variational approximation.**   Using the KL divergence to measure the discrepancy between the target $\pi$ and the pushforward $T_* p$, one can look for a transport map $T$ that minimises the functional

$$T \mapsto \mathcal{D}_{\mathrm{KL}}(T_* \, p \, || \, \pi) \tag{2}$$

over a broad class of functions. The Stein variational method breaks the minimization of (2) into several simple steps: it builds a sequence of transport maps $\{T_1, T_2, \ldots, T_l, \ldots\}$ to iteratively push an initial reference density $p_0$ towards $\pi$. Given a scalar-valued RKHS $\mathcal{H}$ with a positive definite kernel $k(x, x')$, each transport map $T_l : \mathbb{R}^d \to \mathbb{R}^d$ is chosen to be a perturbation of the identity map $I(x) = x$ along the vector-valued RKHS $\mathcal{H}^d \simeq \mathcal{H} \times \cdots \times \mathcal{H}$, i.e.,

$$T_l(x) := I(x) + Q(x) \quad \text{for} \quad Q \in \mathcal{H}^d. \tag{3}$$

The transport maps are computed iteratively. At each iteration $l$, our best approximation of $\pi$ is given by the pushforward density $p_l = (T_l \circ \cdots \circ T_1)_* \, p_0$. The SVGD algorithm then seeks a transport map $T_{l+1} = I + Q$ that further decreases the KL divergence between $(T_{l+1})_* p_l$ and $\pi$,

$$Q \mapsto J_{p_l}[Q] := \mathcal{D}_{\mathrm{KL}}((I + Q)_* \, p_l \,||\, \pi), \tag{4}$$

for an appropriate choice of $Q \in \mathcal{H}^d$. In other words, the SVGD algorithm seeks a map $Q \in \mathcal{H}^d$ such that

$$J_{p_l}[Q] < J_{p_l}[\mathbf{0}], \tag{5}$$

where $\mathbf{0}(x) = 0$ denotes the zero map. By construction, the sequence of pushforward densities $\{p_0, p_1, p_2, \ldots, p_l, \ldots\}$ becomes increasingly closer (in KL divergence) to the target $\pi$. Recent results on the convergence of the SVGD algorithm are presented in [15].

**Functional gradient descent.** The first variation of $J_{p_l}$ at $S \in \mathcal{H}^d$ along $V \in \mathcal{H}^d$ can be defined as

$$DJ_{p_l}[S](V) := \lim_{\tau \to 0} \frac{1}{\tau} \big( J_{p_l}[S + \tau V] - J_{p_l}[S] \big). \tag{6}$$

Assuming that the objective function $J_{p_l} : \mathcal{H}^d \to \mathbb{R}$ is Fréchet differentiable, the *functional gradient* of $J_{p_l}$ at $S \in \mathcal{H}^d$ is the element $\nabla J_{p_l}[S]$ of $\mathcal{H}^d$ such that

$$DJ_{p_l}[S](V) = \langle \nabla J_{p_l}[S], V \rangle_{\mathcal{H}^d} \quad \forall V \in \mathcal{H}^d, \tag{7}$$

where $\langle \cdot, \cdot \rangle_{\mathcal{H}^d}$ denotes an inner product on $\mathcal{H}^d$.

In order to satisfy (5), the SVGD algorithm defines $T_{l+1}$ as a perturbation of the identity map along the steepest descent direction of the functional $J_{p_l}$ evaluated at the zero map, i.e.,

$$T_{l+1} = I - \varepsilon \nabla J_{p_l}[\mathbf{0}], \tag{8}$$

for a small enough $\varepsilon > 0$. It was shown in [17] that the functional gradient at $\mathbf{0}$ has a closed form expression given by

$$-\nabla J_{p_l}[\mathbf{0}](z) = \mathbb{E}_{x \sim p_l}[k(x, z) \nabla_x \log \pi(x) + \nabla_x k(x, z)]. \tag{9}$$

**Empirical approximation.** There are several ways to approximate the expectation in (9). For instance, a set of particles $\{x_i^0\}_{i=1}^n$ can be generated from the initial reference density $p_0$ and pushed forward by the transport maps $\{T_1, T_2, \ldots\}$. The pushforward density $p_l$ can then be approximated by the empirical measure given by the particles $\{x_i^l\}_{i=1}^n$, where $x_i^l = T_l(x_i^{l-1})$ for $i = 1, \ldots, n$, so that

$$-\nabla J_{p_l}[\mathbf{0}](z) \approx G(z) := \frac{1}{n} \sum_{j=1}^n \big[ k(x_j^l, z) \nabla_{x_j^l} \log \pi(x_j^l) + \nabla_{x_j^l} k(x_j^l, z) \big]. \tag{10}$$

The first term in (10) corresponds to a weighted average steepest descent direction of the log-target density $\pi$ with respect to $p_l$. This term is responsible for transporting particles towards high-probability regions of $\pi$. In contrast, the second term can be viewed as a "repulsion force" that spreads the particles along the support of $\pi$, preventing them from collapsing around the mode of $\pi$. The SVGD algorithm is summarised in Algorithm 1.

---

**Algorithm 1:** One iteration of the Stein variational gradient algorithm

---

**Input** : Particles $\{x_i^l\}_{i=1}^n$ at previous iteration $l$; step size $\varepsilon_{l+1}$
**Output** : Particles $\{x_i^{l+1}\}_{i=1}^n$ at new iteration $l+1$
  1: **for** $i = 1, 2, \ldots, n$ **do**
  2:    Set $x_i^{l+1} \leftarrow x_i^l + \varepsilon_{l+1} \, G(x_i^l)$, where $G$ is defined in (10).
  3: **end for**

---

## 3  Stein variational Newton method

Here we propose a new method that incorporates second-order information to accelerate the convergence of the SVGD algorithm. We replace the steepest descent direction in (8) with an approximation of the Newton direction.

**Functional Newton direction.** Given a differentiable objective function $J_{p_l}$, we can define the second variation of $J_{p_l}$ at $\mathbf{0}$ along the pair of directions $V, W \in \mathcal{H}^d$ as

$$D^2 J_{p_l}[\mathbf{0}](V, W) := \lim_{\tau \to 0} \frac{1}{\tau} \big( DJ_{p_l}[\tau W](V) - DJ_{p_l}[\mathbf{0}](V) \big).$$

At each iteration, the Newton method seeks to minimize a local quadratic approximation of $J_{p_l}$. The minimizer $W \in \mathcal{H}^d$ of this quadratic form defines the Newton direction and is characterized by the first order stationarity conditions

$$D^2 J_{p_l}[\mathbf{0}](V, W) = -DJ_{p_l}[\mathbf{0}](V), \quad \forall V \in \mathcal{H}^d. \tag{11}$$

We can then look for a transport map $T_{l+1}$ that is a local perturbation of the identity map along the Newton direction, i.e.,

$$T_{l+1} = I + \varepsilon W, \tag{12}$$

for some $\varepsilon > 0$ that satisfies (5). The function $W$ is guaranteed to be a descent direction if the bilinear form $D^2 J_{p_l}[\mathbf{0}]$ in (11) is positive definite. The following theorem gives an explicit form for $D^2 J_{p_l}[\mathbf{0}]$ and is proven in Appendix.

**Theorem 1.** *The variational characterization of the Newton direction $W = (w_1, \ldots, w_d)^\top \in \mathcal{H}^d$ in* (11) *is equivalent to*

$$\sum_{i=1}^{d} \left\langle \sum_{j=1}^{d} \langle h_{ij}(y, z), w_j(z) \rangle_{\mathcal{H}} + \partial_i J_{p_l}[\mathbf{0}](y), v_i(y) \right\rangle_{\mathcal{H}} = 0, \tag{13}$$

*for all $V = (v_1, \ldots, v_d)^\top \in \mathcal{H}^d$, where*

$$h_{ij}(y, z) = \mathbb{E}_{x \sim p_l} \big[ -\partial_{ij}^2 \log \pi(x) k(x, y) k(x, z) + \partial_i k(x, y) \partial_j k(x, z) \big]. \tag{14}$$

We propose a Galerkin approximation of (13). Let $(x_k)_{k=1}^n$ be an ensemble of particles distributed according to $p_l(\,\cdot\,)$, and define the finite dimensional linear space $\mathcal{H}_n^d = \text{span}\{k(x_1, \cdot), \ldots, k(x_n, \cdot)\}$. We look for an approximate solution $W = (w_1, \ldots, w_d)^\top$ in $\mathcal{H}_n^d$—i.e.,

$$w_j(z) = \sum_{k=1}^{n} \alpha_j^k k(x_k, z) \tag{15}$$

for some unknown coefficients $(\alpha_j^k)$—such that the residual of (13) is orthogonal to $\mathcal{H}_n^d$. The following corollary gives an explicit characterization of the Galerkin solution and is proven in the Appendix.

**Corollary 1.** *The coefficients $(\alpha_j^k)$ are given by the solution of the linear system*

$$\sum_{k=1}^{n} H^{s,k} \alpha^k = \nabla J^s, \quad \text{for all} \quad s = 1, \ldots, n, \tag{16}$$

*where $\alpha^k := \big(\alpha_1^k, \ldots, \alpha_d^k\big)^\top$ is a vector of unknown coefficients, $(H^{s,k})_{ij} := h_{ij}(x_s, x_k)$ is the evaluation of the symmetric form* (14) *at pairs of particles, and where $\nabla J^s := -\nabla J_{p_l}[\mathbf{0}](x_s)$ represents the evaluation of the first variation at the $s$-th particle.*

In practice, we can only evaluate a Monte Carlo approximation of $H^{s,k}$ and $\nabla J^s$ in (16) using the ensemble $(x_k)_{k=1}^n$.

**Inexact Newton.** The solution of (16) by means of direct solvers might be impractical for problems with a large number of particles $n$ or high parameter dimension $d$, since it is a linear system with $nd$ unknowns. Moreover, the solution of (16) might not lead to a descent direction (e.g., when $\pi$ is not log-concave). We address these issues by deploying two well-established techniques in nonlinear optimisation [31]. In the first approach, we solve (16) using the inexact Newton–conjugate gradient (NCG) method [31, Chapters 5 and 7], wherein a descent direction can be guaranteed by appropriately
the matrix-vector product with each $H^{s,k}$ and does not construct the matrix explicitly, and thus can be scaled to high dimensions. In the second approach, we simplify the problem further by taking a

block-diagonal approximation of the second variation, breaking (16) into $n$ decoupled $d \times d$ linear systems

$$H^{s,s}\alpha^s = \nabla J^s, \qquad s = 1, \ldots n \,. \tag{17}$$

Here, we can either employ a Gauss-Newton approximation of the Hessian $\nabla^2 \log \pi$ in $H^{s,s}$ or again use inexact Newton–CG, to guarantee that the approximation of the Newton direction is a descent direction.

Both the block-diagonal approximation and inexact NCG are more efficient than solving for the full Newton direction (16). In addition, the block-diagonal form (17) can be solved in parallel for each of the blocks, and hence it may best suit high-dimensional applications and/or large numbers of particles. In the supplementary material, we provide a comparison of these approaches on various examples. Both approaches provide similar progress per SVN iteration compared to the full Newton direction.

Leveraging second-order information provides a natural scaling for the step size, i.e., $\varepsilon = O(1)$. Here, the choice $\varepsilon = 1$ performs reasonably well in our numerical experiments (Section 5 and the Appendix). In future work, we will refine our strategy by considering either a line search or a trust region step. The resulting Stein variational Newton method is summarised in Algorithm 2.

---

**Algorithm 2:** One iteration of the Stein variational Newton algorithm

---

**Input**  :Particles $\{x_i^l\}_{i=1}^n$ at stage $l$; step size $\varepsilon$
**Output** :Particles $\{x_i^{l+1}\}_{i=1}^n$ at stage $l+1$
  1: **for** $i = 1, 2, \ldots, n$ **do**
  2:    Solve the linear system (16) for $\alpha^1, \ldots, \alpha^n$
  3:    Set $x_i^{l+1} \leftarrow x_i^l + \varepsilon W(x_i^l)$ given $\alpha^1, \ldots, \alpha^n$
  4: **end for**

---

## 4    Scaled Hessian kernel

In the Stein variational method, the kernel weighs the contribution of each particle to a locally *averaged* steepest descent direction of the target distribution, and it also spreads the particles along the support of the target distribution. Thus it is essential to choose a kernel that can capture the underlying geometry of the target distribution, so the particles can traverse the support of the target distribution efficiently. To this end, we can use the curvature information characterised by the Hessian of the logarithm of the target density to design anisotropic kernels.

Consider a positive definite matrix $A(x)$ that approximates the local Hessian of the negative logarithm of the target density, i.e., $A(x) \approx -\nabla_x^2 \log \pi(x)$. We introduce the metric

$$M_\pi := \mathbb{E}_{x \sim \pi}[A(x)] \,, \tag{18}$$

to characterise the average curvature of the target density, stretching and compressing the parameter space in different directions. There are a number of computationally efficient ways to evaluate such an $A(x)$—for example, the generalised eigenvalue approach in [20] and the Fisher information-based approach in [11]. The expectation in (18) is taken against the target density $\pi$, and thus cannot be directly computed. Utilising the ensemble $\{x_i^l\}_{i=1}^n$ in each iteration, we introduce an alternative metric

$$M_{p_l} := \frac{1}{n} \sum_{i=1}^n A(x_i^l), \tag{19}$$

to approximate $M_\pi$. Similar approximations have also been introduced in the context of dimension reduction for statistical inverse problems; see [7]. Note that the computation of the metric (19) does not incur extra computational cost, as we already calculated (approximations to) $\nabla_x^2 \log \pi(x)$ at each particle in the Newton update.

Given a kernel of the generic form $k(x, x') = f(\|x - x'\|^2)$, we can then use the metric $M_{p_l}$ to define an anisotropic kernel

$$k_l(x, x') = f\left( \frac{1}{g(d)} \|x - x'\|_{M_{p_l}}^2 \right),$$

where the norm $\| \cdot \|_{M_{p_l}}$ is defined as $\|x\|_{M_{p_l}}^2 = x^\top M_{p_l} x$ and $g(d)$ is a positive and real-valued function of the dimension $d$. For example, with $g(d) = d$, the Gaussian kernel used in the SVGD of

[17] can be modified as

$$k_l(x, x') := \exp\left(-\frac{1}{2d}\|x - x'\|^2_{M_{p_l}}\right). \tag{20}$$

The metric $M_{p_l}$ induces a deformed geometry in the parameter space: distance is greater along directions where the (average) curvature is large. This geometry directly affects how particles in SVGD or SVN flow—by shaping the locally-averaged gradients and the "repulsion force" among the particles—and tends to spread them more effectively over the high-probability regions of $\pi$.

The dimension-dependent scaling factor $g(d)$ plays an important role in high dimensional problems. Consider a sequence of target densities that converges to a limit as the dimension of the parameter space increases. For example, in the context of Bayesian inference on function spaces, e.g., [26], the posterior density is often defined on a discretisation of a function space, whose dimensionality increases as the discretisation is refined. In this case, the $g(d)$-weighed norm $\|\cdot\|^2/d$ is the square of the discretised $L^2$ norm under certain technical conditions (e.g., the examples in Section 5.2 and the Appendix) and converges to the functional $L^2$ norm as $d \to \infty$. With an appropriate scaling $g(d)$, the kernel may thus exhibit robust behaviour with respect to discretisation if the target distribution has appropriate infinite-dimensional limits. For high-dimensional target distributions that do not have a well-defined limit with increasing dimension, an appropriately chosen scaling function $g(d)$ can still improve the ability of the kernel to discriminate inter-particle distances. Further numerical investigation of this effect is presented in the Appendix.

## 5    Test cases

We evaluate our new SVN method with the scaled Hessian kernel on a set of test cases drawn from various Bayesian inference tasks. For these test cases, the target density $\pi$ is the (unnormalised) posterior density. We assume the prior distributions are Gaussian, that is, $\pi_0(x) = \mathcal{N}(m_{\mathrm{pr}}, C_{\mathrm{pr}})$, where $m_{\mathrm{pr}} \in \mathbb{R}^d$ and $C_{\mathrm{pr}} \in \mathbb{R}^{d \times d}$ are the prior mean and prior covariance, respectively. Also, we assume there exists a forward operator $\mathcal{F} : \mathbb{R}^d \to \mathbb{R}^m$ mapping from the parameter space to the data space. The relationship between the observed data and unknown parameters can be expressed as $y = \mathcal{F}(x) + \xi$, where $\xi \sim \mathcal{N}(0, \sigma^2 I)$ is the measurement error and $I$ is the identity matrix. This relationship defines the likelihood function $\mathcal{L}(y|x) = \mathcal{N}(\mathcal{F}(x), \sigma^2 I)$ and the (unnormalised) posterior density $\pi(x) \propto \pi_0(x)\mathcal{L}(y|x)$.

We will compare the performance of SVN and SVGD, both with the scaled Hessian kernel (20) and the heuristically-scaled isotropic kernel used in [17]. We refer to these algorithms as SVN-H, SVN-I, SVGD-H, and SVGD-I, where 'H' or 'I' designate the Hessian or isotropic kernel, respectively. Recall that the heuristic used in the '-I' algorithms involves a scaling factor based on the number of particles $n$ and the median pairwise distance between particles [17]. Here we present two test cases, one multi-modal and the other high-dimensional. In the supplementary material, we report on additional tests. First, we evaluate the performance of SVN-H with different Hessian approximations: the exact Hessian (full Newton), the block diagonal Hessian, and a Newton–CG version of the algorithm with exact Hessian. Second, we provide a performance comparison between SVGD and SVN on a high-dimensional Bayesian neural network. Finally, we provide further numerical investigations of the dimension-scalability of our scaled kernel.

### 5.1    Two-dimensional double banana

The first test case is a two-dimensional bimodal and "banana" shaped posterior density. The prior is a standard multivariate Gaussian, i.e., $m_{\mathrm{pr}} = 0$ and $C_{\mathrm{pr}} = I$, and the observational error has standard deviation $\sigma = 0.3$. The forward operator is taken to be a scalar logarithmic Rosenbrock function, i.e.,

$$\mathcal{F}(x) = \log\left((1 - x_1)^2 + 100(x_2 - x_1^2)^2\right),$$

where $x = (x_1, x_2)$. We take a single observation $y = \mathcal{F}(x_{\mathrm{true}}) + \xi$, with $x_{\mathrm{true}}$ being a random variable drawn from the prior and $\xi \sim \mathcal{N}(0, \sigma^2 I)$.

Figure 1 summarises the outputs of four algorithms at selected iteration numbers, each with $n = 1000$ particles initially sampled from the prior $\pi_0$. The rows of Figure 1 correspond to the choice of algorithms and the columns of Figure 1 correspond to the outputs at different iteration numbers. We run 10, 50, and 100 iterations of SVN-H. To make a fair comparison, we rescale the number

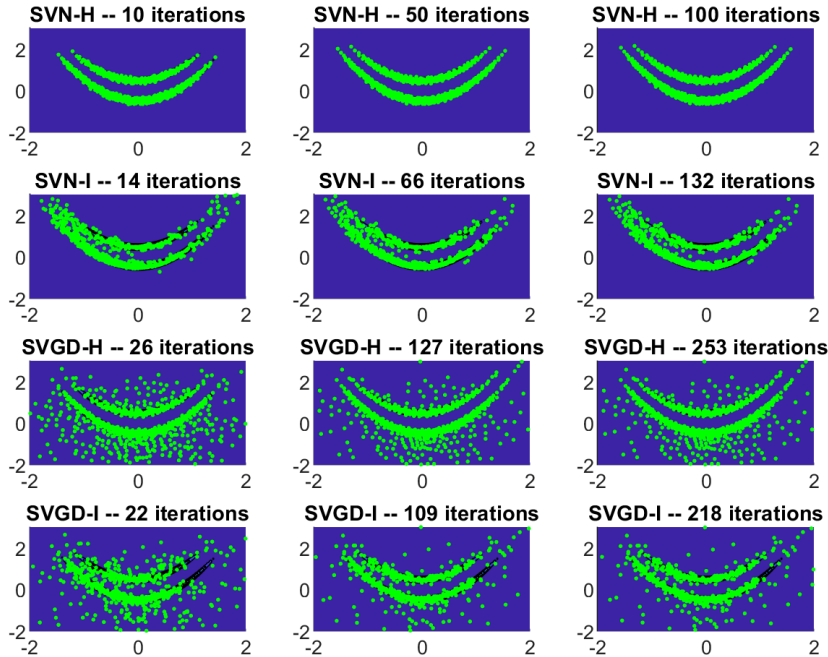

Figure 1: Particle configurations superimposed on contour plots of the double-banana density.

of iterations for each of the other algorithms so that the total cost (CPU time) is approximately the same. It is interesting to note that the Hessian kernel takes considerably less computational time than the Isotropic kernel. This is because, whereas the Hessian kernel is automatically scaled, the Isotropic kernel calculates the distance between the particles at each iterations to heuristically rescale the kernel.

The first row of Figure 1 displays the performance of SVN-H, where second-order information is exploited both in the optimisation and in the kernel. After only 10 iterations, the algorithm has already converged, and the configuration of particles does not visibly change afterwards. Here, all the particles quickly reach the high probability regions of the posterior distribution, due to the Newton acceleration in the optimisation. Additionally, the scaled Hessian kernel seems to spread the particles into a structured and precise configuration.

The second row shows the performance of SVN-I, where the second-order information is used exclusively in the optimisation. We can see the particles quickly moving towards the high-probability regions, but the configuration is much less structured. After 47 iterations, the algorithm has essentially converged, but the configuration of the particles is noticeably rougher than that of SVN-H.

SVGD-H in the third row exploits second-order information exclusively in the kernel. Compared to SVN-I, the particles spread more quickly over the support of the posterior, but not all the particles reach the high probability regions, due to slower convergence of the optimisation. The fourth row shows the original algorithm, SVGD-I. The algorithm lacks both of the benefits of second-order information: with slower convergence and a more haphazard particle distribution, it appears less efficient for reconstructing the posterior distribution.

## 5.2 100-dimensional conditioned diffusion

The second test case is a high-dimensional model arising from a Langevin SDE, with state $u$ : $[0, T] \rightarrow \mathbb{R}$ and dynamics given by

$$du_t = \frac{\beta u \left(1 - u^2\right)}{\left(1 + u^2\right)} \, dt + dx_t, \quad u_0 = 0 \,. \tag{21}$$

Here $x = (x_t)_{t\geq 0}$ is a Brownian motion, so that $x \sim \pi_0 = \mathcal{N}(0, C)$, where $C(t, t') = \min(t, t')$. This system represents the motion of a particle with negligible mass trapped in an energy potential, with thermal fluctuations represented by the Brownian forcing; it is often used as a test case for MCMC algorithms in high dimensions [6]. Here we use $\beta = 10$ and $T = 1$. Our goal is to infer the driving process $x$ and hence its pushforward to the state $u$.

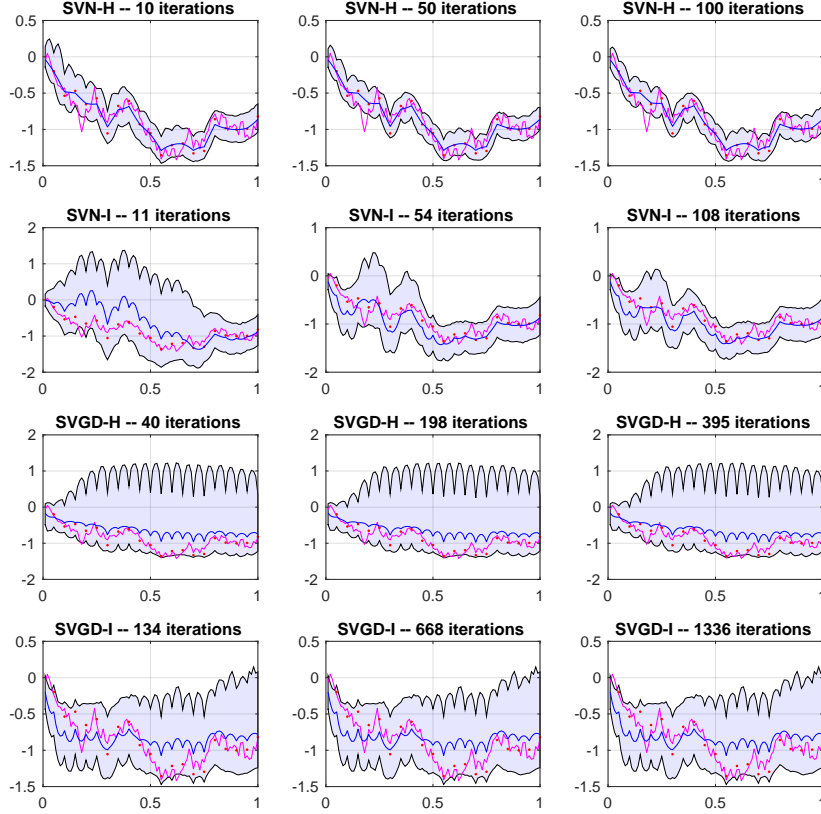

Figure 2: In each plot, the magenta path is the true solution of the discretised Langevin SDE; the blue line is the reconstructed posterior mean; the shaded area is the 90% marginal posterior credible interval at each time step.

The forward operator is defined by $\mathcal{F}(x) = [u_{t_1}, u_{t_2}, \ldots, u_{t_{20}}]^\top \in \mathbb{R}^{20}$, where $t_i$ are equispaced observation times in the interval $(0, 1]$, i.e., $t_i = 0.05\,i$. By taking $\sigma = 0.1$, we define an observation $y = \mathcal{F}(x_{\text{true}}) + \xi \in \mathbb{R}^{20}$, where $x_{\text{true}}$ is a Brownian motion path and $\xi \sim \mathcal{N}(0, \sigma^2 I)$. For discretization, we use an Euler-Maruyama scheme with step size $\Delta t = 10^{-2}$; therefore the dimensionality of the problem is $d = 100$. The prior is given by the Brownian motion $x = (x_t)_{t\geq 0}$, described above.

Figure 2 summarises the outputs of four algorithms, each with $n = 1000$ particles initially sampled from $\pi_0$. Figure 2 is presented in the same way as Figure 1 from the first test case. The iteration numbers are scaled, so that we can compare outputs generated by various algorithms using approximately the same amount of CPU time. In Figure 2, the path in magenta corresponds to the solution of the Langevin SDE in (21) driven by the true Brownian path $x_{\text{true}}$. The red points correspond to the 20 noisy observations. The blue path is the reconstruction of the magenta path, i.e., it corresponds to the solution of the Langevin SDE driven by the *posterior mean* of $(x_t)_{t\geq 0}$. Finally, the shaded area represents the marginal 90% credible interval of each dimension (i.e., at each time step) of the posterior distribution of $u$.

We observe excellent performance of SVN-H. After 50 iterations, the algorithm has already converged, accurately reconstructing the posterior mean (which in turn captures the trends of the true path) and the posterior credible intervals. (See Figure 3 and below for a validation of these results against a reference MCMC simulation.) SVN-I manages to provide a reasonable reconstruction of the

target distribution: the posterior mean shows fair agreement with the true solution, but the credible intervals are slightly overestimated, compared to SVN-H and the reference MCMC. The overestimated credible interval may be due to the poor dimension scaling of the isotropic kernel used by SVN-I. With the same amount of computational effort, SVGD-H and SVGD-I cannot reconstruct the posterior distribution: both the posterior mean and the posterior credible intervals depart significantly from their true values.

In Figure 3, we compare the posterior distribution approximated with SVN-H (using $n = 1000$ particles and 100 iterations) to that obtained with a reference MCMC run (using the DILI algorithm of [6] with an effective sample size of $10^5$), showing an overall good agreement. The thick blue and green paths correspond to the posterior means estimated by SVN-H and MCMC, respectively. The blue and green shaded areas represent the marginal 90% credible intervals (at each time step) produced by SVN-H and MCMC. In this example, the posterior mean of SVN-H matches that of MCMC quite closely, and both are comparable to the data-generating path (thick magenta line). (The posterior means are much smoother than the true path, which is to be expected.) The estimated credible intervals of SVN-H and MCMC also match fairly well along the entire path of the SDE.

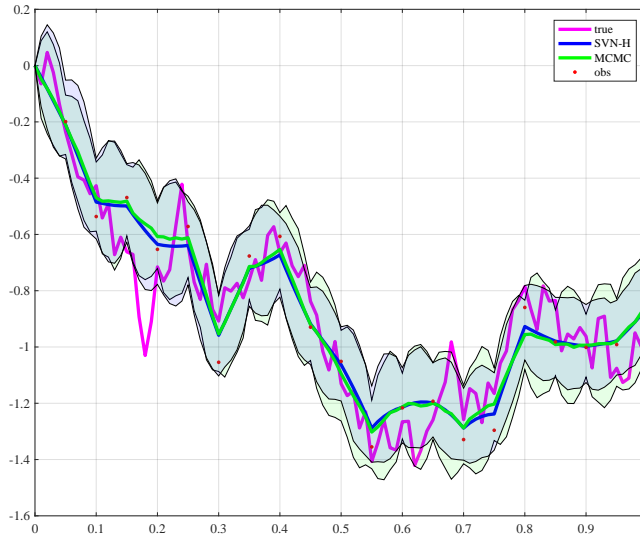

Figure 3: Comparison of reconstructed distributions from SVN-H and MCMC

# 6 Discussion

In general, the use of Gaussian reproducing kernels may be problematic in high dimensions, due to the locality of the kernel [8]. While we observe in Section 4 that using a properly rescaled Gaussian kernel can improve the performance of the SVN method in high dimensions, we also believe that a truly general purpose nonparametric algorithm using local kernels will inevitably face further challenges in high-dimensional settings. A sensible approach to coping with high dimensionality is also to design algorithms that can detect and exploit essential *structure* in the target distribution, whether it be decaying correlation, conditional independence, low rank, multiple scales, and so on. See [25, 29] for recent efforts in this direction.

# 7 Acknowledgements

G. Detommaso is supported by the EPSRC Centre for Doctoral Training in Statistical Applied Mathematics at Bath (EP/L015684/1) and by a scholarship from the Alan Turing Institute. T. Cui, G. Detommaso, A. Spantini, and Y. Marzouk acknowledge support from the MATRIX Program on "Computational Inverse Problems" held at the MATRIX Institute, Australia, where this joint collaboration was initiated. A. Spantini and Y. Marzouk also acknowledge support from the AFOSR Computational Mathematics Program.

## Footnotes

[1] If $T$ is invertible, then $T_* p(x) = p(T^{-1}(x)) \, | \det(\nabla_x T^{-1}(x))|$.

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
