[Supplementary Material]

# Supplementary Material for "A Stein variational Newton method"

**Gianluca Detommaso**
University of Bath & The Alan Turing Institute
gd391@bath.ac.uk

**Tiangang Cui**
Monash University
Tiangang.Cui@monash.edu

**Alessio Spantini**
Massachusetts Institute of Technology
spantini@mit.edu

**Youssef Marzouk**
Massachusetts Institute of Technology
ymarz@mit.edu

**Robert Scheichl**
Heidelberg University
r.scheichl@uni-heidelberg.de

## A  Proof of Theorem 1

The following proposition is used to prove Theorem 1.

**Proposition 1.** *Define the directional derivative of $J_p$ as the first variation of $J_p$ at $S \in \mathcal{H}^d$ along a direction $V \in \mathcal{H}^d$,*

$$DJ_p[S](V) := \lim_{\tau \to 0} \frac{1}{\tau} \big( J_p[S + \tau V] - J_p[S] \big).$$

*The first variation takes the form*

$$DJ_p[S](V) = -\mathbb{E}_{x \sim p} \left[ \big( \nabla_x \log \pi(x + S(x)) \big)^\top V(x) + \text{trace}\big( (I + \nabla_x S(x))^{-1} \nabla_x V(x) \big) \right]. \quad (1)$$

*Proof.* Given the identity map $I$ and a transport map in the form of $T = I + S + \tau V$, the pullback density of $\pi$ is defined as

$$T^* \pi = \pi(T(x)) \, |\det \nabla_x T(x)| = \pi \big( x + S(x) + \tau V(x) \big) \, \big| \det \big( I + \nabla_x S(x) + \tau \nabla_x T(x) \big) \big|.$$

The perturbed objective function $J_p[S + \tau V]$ takes the form

$$
\begin{aligned}
J_p[S + \tau V] &= \mathcal{D}_{\text{KL}}((I + S + \tau V)_* p \, \| \, \pi) \\
&= \mathcal{D}_{\text{KL}}(p \, \| \, (I + S + \tau V)^* \pi) \\
&= \int p(x) \log p(x) dx - \int p(x) \Big( \log \pi \big( x + S(x) + \tau V(x) \big) \\
&\qquad + \log \big| \det \big( I + \nabla_x S(x) + \tau \nabla_x V(x) \big) \big| \Big) dx.
\end{aligned}
$$

Thus we have

$$
\begin{aligned}
J_p[S + \tau V] - J_p[S] = &-\int p(x) \bigg( \underbrace{\log \pi \big( x + S(x) + \tau V(x) \big) - \log \pi(x + S(x))}_{(i)} \bigg) dx \\
&- \int p(x) \big( \underbrace{\log \big| \det \big( I + \nabla_x S(x) + \tau \nabla_x V(x) \big) \big| - \log \big| \det \big( I + \nabla_x S(x) \big) \big|}_{(ii)} \big) dx.
\end{aligned}
$$

$$(2)$$

Performing a Taylor expansion of the terms (i) and (ii) in (2), we have

$$(i) = \tau \big(\nabla_x \log \pi(x + S(x))\big)^\top V(x) + O(\tau^2)\,,$$
$$(ii) = \tau \operatorname{trace}\big((I + \nabla_x S(x))^{-1} \nabla_x V(x)\big) + O(\tau^2)\,,$$

where $\nabla_x \log \pi(x + S(x))$ is the partial derivative of $\log \pi$ evaluated at $x + S(x)$. Plugging the above expression into (2) and the definition of the directional derivative, we obtain

$$DJ_p[S](V) = -\mathbb{E}_{x\sim p}\left[\big(\nabla_x \log \pi(x + S(x))\big)^\top V(x) + \operatorname{trace}\big(\nabla_x(x + \nabla_x S(x))^{-1}\nabla_x V(x)\big)\right].$$
$$(3)$$
$$\square$$

The Fréchet derivative of $J_p$ evaluated at $S \in \mathcal{H}^d$, $\nabla J_p[S] : \mathcal{H}^d \to \mathcal{L}(\mathcal{H}^d, \mathbb{R})$ satisfies

$$DJ_p[S](V) = \langle \nabla J_p[S], V\rangle_{\mathcal{H}^d}, \quad \forall V \in \mathcal{H}^d\,,$$

and thus we can use Proposition 1 to prove Theorem 1.

*Proof of Theorem 1.* The second variation of $J_p$ at $\mathbf{0}$ along directions $V, W \in \mathcal{H}^d$ takes the form

$$D^2 J_p[\mathbf{0}](V, W) := \lim_{\tau\to0} \frac{1}{\tau}\big(DJ_p[\tau W](V) - DJ_p[\mathbf{0}](V)\big)\,.$$

Following Proposition 3, we have

$$D^2 J_p[\mathbf{0}](V, W) = \lim_{\tau\to0} \frac{1}{\tau}\big(DJ_p[\tau W](V) - DJ_p[\mathbf{0}](V)\big)$$
$$= -\mathbb{E}_{x\sim p}\Big[\underbrace{\lim_{\tau\to0}\frac{1}{\tau}\big(\nabla_x \log \pi(x + \tau W(x)) - \nabla_x \log \pi(x)\big)^\top V(x)}_{(i)}\Big]$$
$$- \mathbb{E}_{x\sim p}\Big[\operatorname{trace}\Big(\underbrace{\lim_{\tau\to0}\frac{1}{\tau}[(I + \tau\nabla_x W(x))^{-1} - I]}_{(ii)}\nabla_x V(x)\Big)\Big]\,. \tag{4}$$

By Taylor expansion, the limits (i) and (ii) of the above equation can be written as

$$(i) = \nabla_x^2 \log \pi(x)W(x)\,,$$
$$(ii) = -\nabla_x W(x)\,.$$

Thus, the second variation of $J_p$ at $\mathbf{0}$ along directions $V, W \in \mathcal{H}^d$ becomes

$$D^2 J_p[\mathbf{0}](V, W) = -\mathbb{E}_{x\sim p}\big[W(x)^\top \nabla_x^2 \log \pi(x)V(x) - \operatorname{trace}\big(\nabla_x W(x)\nabla_x V(x)\big)\big]\,. \tag{5}$$

Using the reproducing property of $V, W \in \mathcal{H}^d$, i.e.

$$v_i(x) = \langle k(x, \cdot), v_i(\cdot)\rangle_{\mathcal{H}}\,, \qquad\qquad w_j(x) = \langle k(x, \cdot), w_j(\cdot)\rangle_{\mathcal{H}}$$
$$\nabla_x v_i(x) = \langle \nabla_x k(x, \cdot), v_i(\cdot)\rangle_{\mathcal{H}^d}\,, \qquad \nabla_x w_i(x) = \langle \nabla_x k(x, \cdot), w_i(\cdot)\rangle_{\mathcal{H}^d}$$

we then have

$$\mathbb{E}_{x\sim p}\big[W(x)^\top \nabla_x^2 \log \pi(x)V(x)\big] = \sum_{i=1}^d \sum_{j=1}^d \Big\langle \langle \mathbb{E}_{x\sim p}\big[\partial_{ij}^2 \log \pi(x)k(x, y)k(x, z)\big], w_j(z)\rangle_{\mathcal{H}}, v_i(y)\Big\rangle_{\mathcal{H}}$$

and

$$\mathbb{E}_{x\sim p}\big[\operatorname{trace}\big(\nabla_x W(x)\nabla_x V(x)\big)\big] = \sum_{i=1}^d \sum_{j=1}^d \Big\langle \langle \mathbb{E}_{x\sim p}\big[\partial_i k(x, y)\partial_j k(x, z)\big], w_j(z)\rangle_{\mathcal{H}}, v_i(y)\Big\rangle_{\mathcal{H}}\,.$$

Plugging the above identities into (5), the second variation can be expressed as

$$D^2 J_p[\mathbf{0}](V, W) = \sum_{i=1}^d \sum_{j=1}^d \Big\langle \langle h_{ij}(y, z), w_j(z)\rangle_{\mathcal{H}}, v_i(y)\Big\rangle_{\mathcal{H}}\,,$$

where

$$h_{ij}(y, z) := \mathbb{E}_{x\sim p}\big[-\partial_{ij}^2 \log \pi(x)k(x, y)k(x, z) + \partial_i k(x, y)\partial_j k(x, z)\big]\,.$$

Hence the result. $\square$

# B  Proof of Corollary 1

*Proof.* Here we drop the subscript $p_l$. The ensemble of particles $(x_k)_{k=1}^n$ defines a linear function space $\mathcal{H}_n = \mathrm{span}\{k(x_1, \cdot), \ldots, k(x_n, \cdot)\}$. In the Galerkin approach, we seek a solution $W = (w_1, \ldots, w_d)^\top \in \mathcal{H}_n^d$ such that the residual of the Newton direction

$$\sum_{i=1}^d \left\langle \sum_{j=1}^d \langle h_{ij}(y,z), w_j(z) \rangle_{\mathcal{H}} + \partial_i J[\mathbf{0}](y), v_i(y) \right\rangle_{\mathcal{H}} = 0, \tag{6}$$

is zero for all possible $V \in \mathcal{H}_n^d$. This way, we can approximate each component $w_j$ of the function $W$ as

$$w_j(z) = \sum_{k=1}^n \alpha_j^k \, k(x_k, z), \tag{7}$$

for a collection of unknown coefficients $(\alpha_j^k)$. We define $V^s = (v_1^s, \ldots, v_d^s)^\top$ to be the test function where $v_i^s(y) = k(x_s, y)$ for all $s = 1, \ldots, n$.

We first project the Newton direction (6) onto $V^s$ for all $s = 1, \ldots, n$. Applying the reproducing property of the kernel, this leads to

$$\sum_{j=1}^d \langle h_{ij}(x_s, z), w_j(z) \rangle_{\mathcal{H}^d} + \partial_i J_{p_l}[\mathbf{0}](x_s) = 0, \qquad i = 1, \ldots, d, \quad s = 1, \ldots, n. \tag{8}$$

Plugging (7) into (8), we obtain the fully discrete set of equations

$$\sum_{j=1}^d \sum_{\ell=1}^n h_{ij}(x_s, x_k) \, \alpha_j^k + \partial_i J_{p_l}[\mathbf{0}](x_s) = 0, \quad i = 1, \ldots, d, \ s = 1, \ldots, n, \ k = 1, \ldots, n. \tag{9}$$

We denote the coefficient vector $\alpha^k := \left( \alpha_1^k, \ldots, \alpha_d^k \right)^\top$ for each $x_k$, the block Hessian matrix $(H^{s,k})_{ij} := h_{ij}(x_s, x_k)$ for each pair of $x_s$ and $x_k$, and $\nabla J^s := \nabla J[\mathbf{0}](x_s)$ for each $x_s$. Then equation (9) can be expressed as

$$\sum_{k=1}^n H^{s,k} \, \alpha^k = \nabla J^s, \qquad s = 1, \ldots, n. \tag{10}$$

$\square$

# C  Additional test cases

## C.1  Comparison between the full and inexact Newton methods

Here we compare three different Stein variational Newton methods: `SVNfull` denotes the method that solves the fully coupled Newton system in equation (16) of the main paper, with no approximations; `SVNCG` denotes the method that applies inexact Newton–CG to the fully coupled system (16); and `SVNbd` employs the block-diagonal approximation given in equation (17) of the main paper.

We first make comparisons using the two-dimensional double banana distribution presented in Section 5.1. We run our test case for $N = 100$ particles and 20 iterations. Figure 1 shows the contours of the target density and the samples produced by each of the three algorithms. Compared to the full Newton method, both the block-diagonal approximation and the inexact Newton–CG generate results of similar quality.

We use an additional nonlinear regression test case for further comparisons. In this case, the forward operator is given by

$$\mathcal{F}(x) = c_1 x_1^3 + c_2 x_2 \,,$$

where $x = [x_1, x_2]^\top$ and $c_1, c_2$ are some fixed coefficients sampled independently from a standard normal distribution. A data point is then given by $y = \mathcal{F}(x) + \varepsilon$, where $\varepsilon \sim N(0, \sigma^2)$ and $\sigma = 0.3$. We use a standard normal prior distribution on $x$.

Figure 1: Double-banana example: performance comparison between `SVNfull`, `SVNCG`, and `SVNbd` after 20 iterations

Figure 2: Nonlinear regression example: performance comparison between `SVNfull`, `SVNCG`, and `SVNbd` after 20 iterations

We run our test case for $N = 100$ particles and 20 iterations. Figure 2 shows contours of the posterior density and the samples produced by each of the three algorithms. Again, both the block-diagonal approximation and the inexact Newton–CG generate results of similar quality to those of the full Newton method.

These numerical results suggest that the block-diagonal approximation and the inexact Newton–CG can be effective methods for iteratively constructing the transport maps in SVN. We will adopt these approximate SVN strategies on large-scale problems, where computing the full Newton direction is not feasible.

## C.2 Bayesian neural network

In this test case, we set up a Bayesian neural network as described in [1]. We use the open-source "yacht hydrodynamics" data set[1] and denote the data by $\mathcal{D} = (x_i, y_i)_{i=1}^M$, where $x_i$ is an input, $y_i$ is the corresponding scalar prediction, and $M = 308$. We divide the data into a training set of $m = 247$ input–prediction pairs and a validation set of $M - m = 61$ additional pairs. For each input, we model the corresponding prediction as

$$y_i = f(x_i, w) + \varepsilon_i \,,$$

where $f$ denotes the neural network with weight vector $w \in \mathbb{R}^d$ and $\varepsilon_i \sim N(0, \gamma^{-1})$ is an additive Gaussian error. The dimension of the weight vector is $d = 2951$. We endow the weights $w$ with independent Gaussian priors, $w \sim N(0, \lambda^{-1} I)$. The inference problem then follows from the likelihood function,

$$\mathcal{L}(\mathcal{D}|w, \gamma) = \left( \frac{\gamma}{2\pi} \right)^{\frac{m}{2}} \exp \left( -\frac{\gamma}{2} \sum_{i=1}^m (f(x, w) - y_i)^2 \right),$$

and the prior density,

$$\pi_0(w|\lambda) = \left( \frac{\lambda}{2\pi} \right)^{\frac{m}{2}} \exp \left( -\frac{\gamma}{2} \sum_{i=1}^m w_j^2 \right),$$

where $\gamma$ and $\lambda$ play the role of hyperparameters.

**Performance comparison of SVN-H with SVGD-I.** We compare SVN-H with the original SVGD-I algorithm on this Bayesian neural network example, with hyperparameters fixed to $\log \lambda = -10$ (which provides a very uninformative prior distribution) and $\log \gamma = 0$. First, we run a line-search with Newton–CG to find the posterior mode $w^*$. Figure 3 shows that neural network predictions at the posterior mode almost perfectly match the validation data. Then, we randomly initialise $n = 30$ particles $(x_i)_{i=1}^n$ around the mode, i.e., by independently drawing $x_i \sim \mathcal{N}(w^*, I)$. As in the previous test cases, we make a fair comparison of SVN-H and SVGD-I by taking 10, 20, and 30 iterations of SVN-H and rescaling the number of iterations of SVGD-I to match the computational costs of the two algorithms. Because this test case is very high-dimensional, rather than storing the entire Hessian matrix and solving the Newton system we use the inexact Newton–CG approach within SVN, which requires only matrix-vector products and yields enormous memory savings. Implementation details can be found in our GitHub repository.

Figure 4 shows distributions of the error on the validation set, as resulting from posterior predictions. To obtain these errors, we use the particle representation of the posterior on the weights $w$ to evaluate posterior predictions on the validation inputs $(x_i)_{i=m+1}^M$. Then we evaluate the error of each of these predictions. The red line represents the mean of these errors at each validation input $x_i$, and the shaded region represents the 90% credible interval of these error distribution. Although both algorithms "work" in the sense of producing errors of small range overall, SVN-H yields distributions of prediction error with smaller means and considerably reduced variances, compared to SVGD-I.

## C.3 Scalability of kernels in high dimensions

**Discretization-invariant posterior distribution.** Here we illustrate the dimension scalability of the scaled Hessian kernel, compared to the isotropic kernel used in [1]. We consider a linear Bayesian inverse problem in a function space setting [3]: the forward operator is a linear functional $\mathcal{F}(x) = \langle \sin(\pi s), x(s) \rangle$, where the function $x$ is defined for $s \in [0, 1]$. The scalar observation $y = \mathcal{F}(x) + \xi$, where $\xi$ is Gaussian with zero mean and standard deviation $\sigma = 0.3$. The prior is a Gaussian measure $\mathcal{N}(0, \mathcal{K}^{-1})$ where $\mathcal{K}$ is the Laplace operator $-x''(s)$, $s \in [0, 1]$, with zero essential boundary conditions.

Discretising this problem with finite differences on a uniform grid with $d$ degrees of freedom, we obtain a Gaussian prior density $\pi_0(x)$ with zero mean and covariance matrix $K^{-1}$, where $K$ is the finite difference approximation of the Laplacian. Let the vector $a$ denote the discretised function $\sin(\pi s), s \in [0, 1]$, and let the corresponding discretised parameter be denoted by $x$ (overloading

Figure 3: Neural network prediction at the posterior mode very closely matches the validation data.

notation for convenience). Then the finite-dimensional forward operator can be written as $\mathcal{F}(x) = a^\top x$. After discretization, the posterior has a Gaussian density of the form $\pi = \mathcal{N}(m_{\text{pos}}, C_{\text{pos}})$, where

$$m_{\text{pos}} = \frac{y}{\sigma^2} C_{\text{pos}}\, a\,, \qquad C_{\text{pos}} = \left(K^{-1} + \frac{1}{\sigma^2} a a^\top\right)^{-1}.$$

To benchmark the performance of various kernels, we construct certain summaries of the posterior distribution. In particular, we use our SVN methods with the scaled Hessian kernel (SVN-H) and the isotropic kernel (SVN-I) to estimate the component-wise average of the posterior mean, $\frac{1}{d}\sum_{i=1}^d m_{\text{pos},i}$, and the trace of the posterior covariance, $\text{trace}(C_{\text{pos}})$, for problems discretised at different resolutions $d \in \{40, 60, 80, 100\}$. We run each experiment with $n = 1000$ particles and 50 iterations of SVN. We compare the numerical estimates of these quantities to the analytically known results. These comparisons are summarised in Tables 1 and 2.

From Table 1, we can observe that all algorithms almost perfectly recover the average of the posterior mean up to the first three significant figures. However, Table 2 shows that SVN-H does a good job in estimating the trace of the posterior covariance consistently for all dimensions, whereas SVN-I under-estimates the trace—suggesting that particles are under-dispersed and not correctly capturing the uncertainty in the parameter $x$. This example suggests that the scaled Hessian kernel can lead to a more accurate posterior reconstruction for high-dimensional distributions than the isotropic kernel.

**A posterior distribution that is not discretization invariant.** Now we examine the dimension-scalability of various kernels in a problem that does not have a well-defined limit with increasing parameter dimension. We modify the linear Bayesian inverse problem introduced above: now the prior covariance is the identity matrix, i.e., $K^{-1} = I$ and the vector $a$ used to define the forward operator is drawn from a uniform distribution, $a_i \sim \mathcal{U}(2, 10),\ i = 1, \ldots, d$. This way, the posterior is not discretization invariant. We perform the same set of numerical experiments as above and summarise the results in Tables 3 and 4. Although the target distribution used in this case is not

Figure 4: Bayesian neural network example: Comparison between SVN-H and SVGD-I, showing the distribution of errors between the validation data and samples from the posterior predictive.

Table 1: Comparison of theoretical and estimated averages of the posterior mean

| Averages of the posterior mean $\frac{1}{d} \sum_{i=1}^{d} m_{\text{pos},i}$ | | | | |
|---|---|---|---|---|
| $d$ | 40 | 60 | 80 | 100 |
| Theoretical | 0.4658 | 0.4634 | 0.4622 | 0.4615 |
| SVN-H | 0.4658 | 0.4634 | 0.4623 | 0.4614 |
| SVN-I | 0.4657 | 0.4633 | 0.4622 | 0.4615 |

Table 2: Comparison of theoretical and estimated traces of the posterior covariance

| Traces of the posterior covariance $\text{trace}(C_{\text{pos}})$ | | | | |
|---|---|---|---|---|
| $d$ | 40 | 60 | 80 | 100 |
| Theoretical | 0.1295 | 0.1297 | 0.1299 | 0.1299 |
| SVN-H | 0.1271 | 0.1281 | 0.1304 | 0.1293 |
| SVN-I | 0.0925 | 0.0925 | 0.0925 | 0.0923 |

discretization invariant, the scaled Hessian kernel is still reasonably effective in reconstructing the target distributions of increasing dimension (according to the summary statistics below), whereas the isotropic kernel under-estimates the target variances for all values of dimension $d$ that we have tested.

## Footnotes

[1] http://archive.ics.uci.edu/ml/datasets/yacht+hydrodynamics

# References

[1] Q. Liu and D. Wang. Stein variational gradient descent: A general purpose Bayesian inference algorithm. In *Advances In Neural Information Processing Systems* (D. D. Lee et al., Eds.),

Table 3: Comparison of theoretical and estimated averages of the posterior mean

| Averages of the posterior mean $\frac{1}{d}\sum_{i=1}^{d} m_{\mathrm{pos},i}$ | | | | |
|---|---|---|---|---|
| $d$ | 40 | 60 | 80 | 100 |
| Theoretical | 0.0037 | 0.0025 | 0.0019 | 0.0015 |
| SVN-H | 0.0037 | 0.0025 | 0.0019 | 0.0015 |
| SVN-I | 0.0037 | 0.0025 | 0.0019 | 0.0015 |

Table 4: Comparison of theoretical and estimated traces of the posterior covariance

| Traces of the posterior covariance trace($C_{\mathrm{pos}}$) | | | | |
|---|---|---|---|---|
| $d$ | 40 | 60 | 80 | 100 |
| Theoretical | 39.0001 | 59.0000 | 79.0000 | 99.0000 |
| SVN-H | 37.7331 | 55.8354 | 73.6383 | 90.7689 |
| SVN-I | 8.7133 | 8.2588 | 7.9862 | 7.6876 |

Vol. 29, p. 2378–2386, 2016.

[2] R. M. Neal *Bayesian learning for neural networks*. Springer - Lecture Notes in Statistics

[3] A. M. Stuart. Inverse problems: a Bayesian perspective. In *Acta Numerica*, 19, p. 451–559, 2010.