[Reviews · NeurIPS 2018]

Reviewer 1



Summary: SVGD iteratively moves a set of particles toward the target by choosing a perturbative direction to maximumly decrease the KL divergence with the target distribution in RKHS. The paper proposes to add second-order information into SVGD updates, preliminary empirical results show that their method converges faster in few cases. The paper is well written, and the proofs seem correct. An important reason in using second-order information is the hope to achieve a faster convergence rate. My major concern is a lack of theoretical analysis of convergence rate in this paper: 1) An appealing property of SVGD is that the optimal decreasing rate equals to Stein discrepancy D_F(q||p), where F is a function set that includes all possible velocity fields. D(q||p) = 0 iff q = p. Taking F as RKHS, a closed form solution of the perturbation direction is derived in [Liu et al., 2016, Chwialkowski et al., 2016]. However, the authors didn’t draw any connections between their updates and Stein discrepancy. It’s not clear at all why SVN would yield a faster convergence rate than the original SVGD. And if so, in what sense that SVN is faster? Would the updates of SVN equal to a sum of Stein discrepancy and some other positive terms? - Liu et al., 2016, A kernelized Stein discrepancy for goodness-of-fit tests - Chwialkowski et al., 2016. A Kernel Test of Goodness of Fit 2) It would also help l if the authors could discuss the convergence properties of their approach in terms of theνmberofpartics. e.g. Liu et al., 2017, Stein Variational Gradient Descent as Gradient Flow. 3) Calculating the hessian-matrix is computationally expensive. For the purpose of efficiency, the authors used few wild approximations in their update, e.g. (line 127-138) the cross term in H_l is dropped, and the second-order term is replaced by its Gauss-Newton approximation, etc. These simplifications could reduce the computational cost, but it’s not clear how they would affect the convergence analysis. 4) In the current examples, comparisons were only done with low dimensions and toy examples. Probably the results would be more convincing if the authors could test with high dimensions, e.g. the Bayesian neural network example.

Reviewer 2



The authors propose an version of Stein variational gradient descent (SVGD) that uses approximate second-order information, both to improve the optimization (by analogy with other quasi-Newton optimization methods) and to create a geometry-aware kernel (by analogy with pre-conditioning). The use of second order information improves convergence considerably over standard SVGD, with the greatest benefit provided by both the geometry-aware kernels and second-order optimization steps. Stein methods for approximate inference are an exciting topic these days, the improvements are both practically feasible and significant, and the paper is very well-written. I will argue that the paper should be accepted and really only have minor comments and suggestions. - A more appropriate title is “A Stein variational Quasi-Newton method”, since the authors are not actually able to take a full Newton step, even on small problems, due to the fact that the transforms, and hence push-forward density, are not available in closed form (line 128). - In general, I might replace the phrase “inexact Newton” with “quasi-Newton” for continuity with other optimiziation literature (e.g. line 130). - “There exist infinitely many such maps” (line 62) is repeated from the introduction. - At line 80, you can define equation (6) without Frechet differentiability; it is equation (7) that requires it (specifically, that \nabla J_{p_l} does not depend on V). So for clarity perhaps it would be better to move the assumption of Frechet differentiability to line 81. - I think it might be better to spell out the definition of N(x) at line 131. I gathered that it is the outer product of the gradient of log p(x) at x, but it would be better to say that explicitly since this is an important part of the algorithm. Calling it equivalent to the Fisher information matrix is confusing -- the Fisher information matrix is usually as the covariance of the scores, whereas I think the authors don’t intend any expectation to be calculated. Furthermore, a correctly specified Fisher information matrix could be calculated either from the scores or from the Hessian, and the authors want to emphasize that it is the former, not the latter. - At line 150, the authors define a matrix, not a metric, so it would be better to call it that. This matrix is later used to /form/ a metric. - At line 236, is it supposed to be F(x), not F(u)? In line 238 F is applied to the Brownian motion, not the solution to the SDE. - Line 254, for comparison I’d be interested to know how long HMC took.

Reviewer 3



Thank you for an interesting read. This paper proposed a newton-like method in function space for approximate Bayesian inference. Specifically, the authors define functional newton direction, and propose Monte Carlo approximation to it. Furthermore the authors also proposes an improvement of the kernel by using the approximately computed second-order information. Experimental results show significant improvement over naive SVGD. I think this paper can be a good contribution to the approximate inference community, however the presentation can be improved. Specifically, the "inexact newton" paragraph needs substantial explanation: 1. Why it is safe to drop the \nabla_x k \nabla_x log p term? 2. I don't understand the justification of approximation (18) to (16). I would also like to see at least a numerical analysis on the error of this inexact Newton approximation. You can construct synthetic densities that you can compute the exact hessian of log \pi(x). Also the experiments, although with significantly better results, seems a bit toy. The original SVGD paper tested the algorithm on Bayesian neural network regression, and the dimensionality there is at least ~300 I believe. ================================================ I have read the feedback. I don't feel the feedback has addressed my concerns on the "inexact Newton" part. While further theoretical analysis might be hard at this point, I would like to see at least a toy example that you can compute exact Newton, and compare it to your approximation method.

Reviewer 4



The authors proposed a new variational inference method -- a Stein variational Newton (SVN) method. It is based on Stein variational gradient decent (SVGD) method developed in [11], and brings two main contributions. The first contribution is to apply a Newton iteration that exploits the second-order information of the probability distribution to push forward the samples. To avoid the challenge of solving a large coupled system arising from the Newton iteration, the authors present two approximations -- one dropping a cross term involving the current density function, the other using a mass lumping" method to decouple the system. The second contribution is the design of the kernel (of the RKHS used in the method) by incorporating the averaged Hessian of the negative logarithm of the target density function. Compared to a Gaussian kernel with isotropic structure, this rescaled kernel carries the curvature information of the target distribution which prevents samples from collapsing in high dimensions. These contributions largely accelerate the convergence of the samples to the target distribution, thus make SVN method computationally more efficient and tractable. The efficiency and tractability are demonstrated by several numerical examples with parameter dimension ranging from low (2) to high (100). Overall, this new development brings important contribution to the family of variational inference methods; the numerical demonstration is very convincing; the paper is easy to read and well-written. I am confident of the clarity and correctness of the algorithmic development. I also run the code and obtain statistically the same results and conclusions. Therefore, I recommend to accept it after some minor revisions. 1. page 5, Algorithm 2: Changing the step size of the Newton iterations looks important for the fast convergence. A few more details on how the author changed the step size is helpful. 2. page 5, line 174, it is not clear under what technical conditions the g(d)-weighted norm is the square of the discretized L2 norm. 3. Can the author explain why the isotropic kernel takes less time compared to the Hessian kernel, as can be observed from the different number of iterations in Figure 1? 4. supplementary material, page 4, after line 48, should the y" be x"?